# Healthcare Associated Infections—A New Pathology in Medical Practice?

**DOI:** 10.3390/ijerph17030760

**Published:** 2020-01-25

**Authors:** Septimiu Voidazan, Sorin Albu, Réka Toth, Bianca Grigorescu, Anca Rachita, Iuliu Moldovan

**Affiliations:** 1Department of Epidemiology, University of Medicine, Pharmacy, Sciences and Technology George Emil Palade of Tîrgu Mureș, 540141 Tîrgu Mureș, Romania; septi_26_07@yahoo.com; 2Department of Physiology, University of Medicine, Pharmacy, Sciences and Technology George Emil Palade of Tîrgu Mureș, 540141 Tîrgu Mureș, Romania; 3Department of Quality Management in Healthcare Services, County Emergency Clinical Hospital of Tîrgu Mureș, 540141 Tîrgu Mureș, Romania; bodeareka@gmail.com; 4Department of Pathophysiology, University of Medicine, Pharmacy, Sciences and Technology George Emil Palade of Targu-Mures, 540141 Tîrgu Mureș, Romania; biancagrigorescu20@yahoo.com; 5University of Medicine, Pharmacy, Sciences and Technology George Emil Palade of Targu-Mures, 540141 Tîrgu Mureș, Romania; anca.rachita@umfst.ro; 6Discipline of public health and health management University of Medicine, Pharmacy, Science and Technology George Emil Palade of Targu-Mures, 540141 Tîrgu Mureș, Romania; iuliu.moldovan@yahoo.com

**Keywords:** hospital acquired infections, prevention and control strategies, bacterial resistance

## Abstract

*Background:* Hospital-acquired infections (HAI) contribute to the emotional stress and functional disorders of the patient and in some cases, can lead to a state of disability that reduces quality of life. Often, HAI are one of the factors that lead to death. The purpose of this study was to analyze the cases of HAI identified in public hospitals at the county level, through case report sheets, as they are reported according to the Romanian legislation. *Methods:* We performed a cross sectional study design based on the case law of the data reported to the Mures Public Health Directorate, by all the public hospitals belonging to this county. We tracked hospital-acquired infections reported for 2017–2018, respectively, a number of 1024 cases, which implies a prevalence rate of 0.44%, 1024/228,782 cases discharged from these hospitals during the studied period. *Results:* The most frequent HAIs were reported by the intensive care units (48.4%), the most common infections being the following: bronchopneumonia (25.3%), enterocolitis with *Clostridioides difficile* (23.3%), sepsis, surgical wound infections and urinary tract infections. At the basis of HAI were 22 pathogens, but the five most common germs were *Clostridioides difficile*, *Acinetobacter baumannii, Klebsiella pneumoniae, Pseudomonas aeruginosa* and *Staphylococcus aureus*. Bronchopneumonia have been most frequently reported in intensive care units, the most common being identified the *Acinetobacter baumannii* agent. Sepsis and central catheter infections also appeared predominantly in intensive care units, more often with *Klebsiella pneumoniae*. The enterocolitis with *Clostridioides difficile*, were the apanage of the medical sections. Infections with *Staphylococcus aureus* have been identified predominantly in the surgical sections at the level of the surgical wounds. Urinary infections had a similar distribution in the intensive care units, the medical and surgical sections, with *Klebsiella pneumoniae* being the most commonly incriminated agent. *Conclusions:* We showed a clear correspondence between the medical units and the type of HAI: what recommends the rapid, vigilant and oriented application of the prevention and control strategies of the HAI.

## 1. Introduction

Hospital acquired infections (HAI) are considered a public health problem, by increasing mortality and morbidity, increasing the length of hospitalization and very high costs for healthcare. In Europe, HAI causes 16 million additional hospitalization days each year, with costs exceeding 7 billion euros annually [1,2,3,4].

The early identification and diagnosis of HAI based on standardized definitions is a first step in the correct management of the cases. For a better understanding of the magnitude of the phenomena and for the rigorous application of appropriate measures, they should always be reported.

HAI prevalence in high-income countries is of 7.5%, although others have reported rates of 5.7%–7.1% in Europe and 4.5% in the US, while in low- and middle-income countries, the prevalence rate ranges between 5.7% and 19.2% [5,6,7,8,9]. The prevalence rate differs from one country to another depending on the possibilities of infection prevention and control [10]. In Europe and North America, it is estimated that 12%–32% of infections associated with blood lead to death. However, the exact burden of HAI in each country is not yet known [11,12].

In Romania, HAI represent a much-underestimated pathology. According to the official reports of HAI communicated by hospitals, the prevalence rates are only of 0.2%–0.25%. More credible data for Romania were identified in a European study in 2018, namely that the percentage is 2.6% (Confidence Interval (CI) 95%: 1.7–4.0), which would represent approx. 100,000 cases registered annually in Romania [13].

Once with the approval of Order no. 1101 from 2016 [14], regarding the approval of the Norms of surveillance, prevention and limitation of the HAI in the health units, in Romania, the reporting rate of HAI has increased, but the prevalence rate is still far from the European average. The Norms of surveillance, prevention and limitation of HAI in the medical units provided in this Order include: (a) organization of the activities of surveillance, prevention and limitation of the HAI in the public and private medical units with beds; (b) the surveillance and the reporting of HAI; (c) methodology for monitoring the accidental exposure of personnel working in the healthcare field to biological products and (d) Standard precautions—mandatory minimum measures for the prevention and limitation of infections associated with healthcare. The detection/identification, registration and declaration/reporting of the HAI by any health unit are mandatory and the surveillance activities, prevention and limitation of the HAI are part of the professional obligations of the personnel and are included in the job description of each employee. As general provisions of this Order at the level of all sanitary units with beds, state or private, specialized Services/Compartments for the prevention of HAI (HAIPS) are organized, which have at least one epidemiologist, with the function/duties of the head of service. Furthermore, in all the sanitary units with beds is set up the HAI Prevention Committee, which includes the head of the HAI prevention department, the doctor responsible for the antibiotic use policy, the medical director, the care director, as the case may be, the pharmacist and the microbiologist.

The purpose of this study was to analyze the cases of HAI identified from the public hospitals at the county level through the case reports sheets as they are reported according to the Romanian legislation.

## 2. Material and Methods

We performed a cross-sectional study design based on the cases reported to the Mures Public Health Directorate, by all the hospitals assigned to Mures County, namely two county clinical hospitals, one cardiovascular disease institute, three municipal hospitals and one city hospital. All these hospitals have a mean average of 3817 beds annually. We mentioned that these hospitals are public; no case was reported from any private hospital. In the county area there are several hospitals and private health units, with continuous hospitalization services or one-day hospitalization, only three of them are larger (up to 200 beds) which have medical and surgical sections. In these hospitals, the patients are usually hospitalized by appointment, sometimes requiring a microbiological control in order to determine the nosocomial risk at the moment of hospitalization.

Mures County is located in the Central Region of Romania with a population of 540,000 inhabitants (about 2.76% of the total population in Romania), with a similar distribution by gender and place of origin. We tracked the HAIs reported for 2017–2018, respectively, a number of 1024 cases, which implies a prevalence rate of 0.44%, namely 1024/228,782 cases discharged from these hospitals during the studied period. According to order 1101/2016, the data related to the case report of HAI are reported to the Mures Public Health Directorate [14].

Thus, we performed a statistical processing according to: hospital, department, specialty, age of the patient, gender, residence, hospitalization diagnosis, diagnosis of HAI, the date and type of surgery or the time of application of a medical device, the date and type of antibiotic administered, if the patient was isolated, if he was in contact with other patients when the infection associated with healthcare was suspected, the patient’s status upon discharge, the date and cause of death, the pathogen involved and the antibiotic resistance in the case of a microbial pathogen. For each case of HAI, the antibiogram was performed. The cases reported there were from 57 *medical sections* (intensive care anesthesia, infectious diseases, cardiology, diabetes, hematology, gastroenterology, palliative care, internal medicine, neurology, neonatology, nephrology, oncology, pediatrics, etc.) and *surgical ones* (general surgery, plastic surgery, obstetric gynecology, orthopedics, neurosurgery, etc.), SMURD (the name is the Romanian acronym for “*Serviciul Mobil de Urgență, Reanimare și Descarcerare*”, which means *Mobile Emergency Service for Resuscitation and Extrication*. These sections were grouped for easier statistical analysis into five categories, namely, intensive care anesthesia (adults, children), surgical units, medical units, neonatology-pediatric units, SMURD.

For processing the data, we received the consent of the Mures Public Health Directorate, we kept the anonymity of the hospitals that reported these infections.

### 2.1. Definition of HAI

An “*infection associated with the medical activities—HAI*” according to Order no. 1101/2016 regarding the approval of the Norms of supervision, prevention and limitation of the infections associated to the medical assistance in the healthcare units), [14] can be defined as *“the infection contracted in sanitary units with beds (state and private) and refers to any infectious disease that can be clinically and/or microbiologically recognized and for which there is epidemiological evidence of contracting it during hospitalization/medical act or medical maneuvers, which affects either the patient due to the medical care received, or the healthcare staff due to his activity, and is linked by incubation to the period of medical care in the respective unit, whether or not the symptoms of the disease appear during the hospitalization period. It must be proved that the infection is due to the hospitalization or the medical-sanitary care in the sanitary units and that it was not in the incubation or in the phase of initiation/clinical evolution at the time of hospitalization/medical act/medical maneuver.”*

### 2.2. Identification and Diagnosis of HAI in Romania

Identification of suspected cases of HAI is done by the attending physician. He notifies the hospital’s HAIPS by phone of each suspected HAI case on the detection day. In order to establish the diagnosis of HAI, the case definitions provided in Decision 2012/506 / EU will be respected [15].

The suspected cases of HAI are investigated with the laboratory and paraclinic within 24 h after the suspicion was raised. The laboratory of medical analyzes, compartment of bacteriology will announce, on the day of identification, the cases in which the multidrug resistant/epidemiological germs were identified at the HAIPS and to the section where the case was suspected.

The classification of cases in confirmed/denied/colonization is done by the attending physician in collaboration with the epidemiologist, by corroborating the clinical, paraclinical data, according to the case definitions.

The HAI case statement is made on the HAI case file by the attending physician who will complete, sign and stamp this form. The diagnosis of HAI will be mentioned by the attending physician in the general clinical observation sheet and in other medical documents, if applicable (e.g., consultation sheet, consultation record, the unique electronic HAI Monitoring Register etc.). At the same time, for the data matching and a correct record, for each HAI case, the diagnosis of nosocomial infection—coded Y 95, in the general clinical observation sheet and in the computer system will be completed at the discharge/transfer of the patient. The responsibility of the correctness of the recorded data rests with the doctor in whose care the patient is. The sections and the compartments with beds will send the unique electronic HAI Monitoring Registry to the HAIPS: daily—if there are suspected cases of HAI, respectively for the confirmed cases and declared on the HAI type sheet (on the reporting day)—weekly. At the centralized hospital level, the HAIPS will conduct the HAI case records throughout the hospital by registering them in the Centralized HAI Record Register.

The HAIPS prepares the epidemiological investigation for each case of HAI reported and confirmed with the measures in place to prevent the nosocomial transmission of pathogens. The HAIPS centralizes data on the cases of HAI diagnosed and reported. Monthly, quarterly and annual reports are prepared on the incidence of nosocomial infections, by sections and types of infections that are submitted to the hospital’s Steering Committee and the Public Health Directorate of the county.

### 2.3. Statistical Analysis

Statistical analysis was performed using the Statistical Package for Social Sciences (SPSS. version 20. Chicago, IL, USA). Quantitative data were presented as mean and standard deviation for normally distributed data (age). Qualitative data were presented as counts and percentages.

## 3. Results

### Participants’ Characteristics

Of the total 1024 cases of HAI, 58.1% were identified in men, 52.9% in the urban area. The average age of the persons identified with HAI was 60.6 ± 19.45 years (mean ± standard deviation), with a minimum of one month and a maximum of 99 years. The most frequent HAI were reported by the sections of ICU, (intensive care unit—48.4%), the most frequent infection: bronchopneumonia (25.3%), followed by the enterocolitis *Clostridioides difficile* (23.3%). Of the reported cases, 25.3% were declared deaths, and data analysis shows that in 86.7% were considered HAI from the reporting hospital (Table 1).

The most common pathogen was *Clostridioides difficile* (22.3%), most commonly found in medical units causing enterocolitis. In descending order, there was *Acinetobacter baumannii* (16.7%), more frequently on intensive care units, *Klebsiella pneumoniae* (14.8%), more frequently on intensive care units, *Pseudomonas aeruginosa* (11.2%), more often on intensive care units, and *Staphylococcus aureus* (10.7%), especially on the surgical sections (Table 1; Figure 1).

Most of the bronchopneumonias had as aetiology the infections with *Acinetobacter baumannii* (60.8%), the surgical wound infections with *Staphylococcus aureus* (41.8%), the urinary tract infection with *Escherichia coli* (29.2%) and *Klebsiella pneumoniae* (28.9%), and sepsis has as main source of infection *Klebsiella pneumoniae* (28.3%) and *Staphylococcus aureus* (23.6%). Other infections included central catheter infection, influenza type B virus, meningitis, external ventricular drainage infection, otitis, phlebitis (Figure 2).

Out of the 1024 reported HAI, for 556 patients, isolation conditions could be created. Nearly one third of them were infected with *Clostridioides difficile*. There were contacts in 702 cases, most of them being infections with *Acinetobacter baumannii* and *Klebsiella pneumoniae* (Figure 3). Deaths were reported in 259 of the cases (25.3%), most of them related to *Acinetobacter baumannii* (39.2%) and *Pseudomonas aeruginosa* (32.2%) infections. In 68 patients (6.6%) healing was possible, or improvement upon discharge (30.3%), but we have to mention that in 320 (31.3%) of the cases, we did not have data specified in the HAI case report to the Public Health Directorate (Table 1; Figure 4). There are also 770 (75.2%) cases without data when the cause of death had to be specified; only in 11 cases (1.12%), the cause of death could be incriminated as a HAI.

## 4. Discussion

This study followed the HAI rate for a 2-year period in a county in which 1024 cases were reported from 7 public hospitals. The average age of the cases was towards the 6–7 decade of age. In addition to this risk factor, many of the people affected had many comorbidities, often multiple hospitalizations with additional use of diagnostic and therapeutic procedures that affect the host’s immune system, the use of antibiotic treatments. Older people, who represent the vast majority of the population in need of medical services, are more vulnerable to infections due to reduced immunological competence, due to multiple morbidities and chronic diseases [16,17].

No cases were reported from private hospitals. A strong point of private hospitals is hygiene. First, they are not as crowded as the public ones, and infectious pathogens do not circulate freely from one patient to another. Secondly, the wards of the contagious people are individual, and this means that the chances of contracting any infectious disease are lower. Moreover, the budget allocated to sanitation in private hospitals is according to the tariffs they charge; the budget allows them to be equipped with the latest medical devices and to permanently ensure the stock of cleaning and disinfection materials. Last but not least, the length of hospitalization is shorter so the risk of infections associated with prolonged hospitalization is highly reduced. The emergence of infections associated with healthcare in these units, would attract negative publicity, and greater chances of being requested material and moral damages by the patient, since he pays any kind of consultation and any procedure. We believe that the non-reporting of infections associated with healthcare due to their lack, from private units, is due to the much greater possibilities of their supervision and prevention.

The HAI rate for the monitored hospitals in our study was of 0.44%. Most cases of HAI have been reported on the ICUs, but it should be noted that many cases are transferred to these sections from other medical or surgical units, as the clinical progression of patients becomes unfavorable, due to the associated comorbidities or complications. Many of these patients get mechanically intubated and ventilated or other medical devices (urethral bladder, peripheral or central venous catheters, drains, etc.) are applied, which are favorable factors that can contribute to the infectious process.

A special feature of these patients is the early impairment of the immune response, due, on the one hand, to chronic disease decompensations, to previous antibiotic or immunosuppressive treatments, to the plastic nutritional status and implicitly, the delay in wound healing. The presence of venous, arterial catheters, as well as the devices required for advanced life support are germ entry gates. Prolonged mechanical ventilation, prolonged use of proton pump inhibitors or prolonged antibiotic therapy are responsible for the onset of ventilator pneumonia or *Clostridioides difficile* infections.

Of the infections reported in our study, a top five for bronchopneumonia, enterocolitis with *Clostridioides difficile*, sepsis, surgical wound infections and urinary tract infections may be established. There are 22 pathogens at the basis of HAI but the five most common germs were *Clostridioides difficile*, *Acinetobacter baumannii*, *Klebsiella pneumoniae*, *Pseudomonas aeruginosa* and *Staphylococcus aureus*.

By analyzing our data, we identified a correspondence between the type of infection, the medical unit and the type of pathogen. Thus, bronchopneumonias have been reported more frequently in intensive care units with *Acinetobacter baumannii*, the most common agent. Sepsis and central catheter infections also appeared predominantly in intensive care units, more often with *Klebsiella pneumoniae*. Enterocolitis with *Clostridioides difficile*, were the apanage of the medical sections. Infections with *Staphylococcus aureus* have been identified mainly in the surgical sections in surgical wounds. The urinary infections had a similar distribution in the intensive care units, the medical and surgical sections, with *Escherichia coli* and *Klebsiella pneumoniae* being the most frequently encountered agents.

According to a European report [10], the percentage of pneumonia or lower respiratory tract infections ranged from 12.0% in Sweden to 36.3% in Lithuania. The percentage of lower tract infections ranged from 10.1% in Cyprus to 30.7% in France. The proportion of surgical wound infections ranged from 8.8% in Luxembourg to 29.0% in Spain. The rates of sepsis were highest in Greece (18.9%) and Cyprus (19.0%) and the lowest in Iceland (2.0%) and were secondary to another infection in 28.8% of cases, (ranging from 0% in Iceland, Latvia and Romania to 40% in Belgium, Denmark, Estonia, Germany, Luxembourg, Malta, Holland, Norway, Slovenia and Sweden). Out of a total of 15,000 reported HAIs, the most common reported types were respiratory tract infections (pneumonia: 19.4%; lower respiratory tract infections: 4.1%), surgical wound infections (19.6%), urinary tract infections (19.0%), blood infections (10.7%) and gastrointestinal infections (7.7%); *Clostridioides difficile* infections were responsible for 48% of all gastrointestinal infections and for 3.6% of all HAI. The ten most common isolated microorganisms were: *Escherichia coli* (15.9%), *Staphylococcus aureus* (12.3%), *Enterococcus* species (9.6%), *Pseudomonas aeruginosa* (8.9%), *Klebsiella* species (8, 7%), negative coagulase staphylococci (7.5%), *Candida* species (6.1%), *Clostridioides difficile* (5.4%), *Enterobacter* species (4.2%), *Proteus* species (3.8%) and *Acinetobacter* species (3.6%).

In M.F. Furmenti’s et al. [18] study, on a group of 957 reported HAI, the most commonly reported HAI were respiratory infections, especially those affecting the lower respiratory tract level (73.7%) and urinary tract infections 46.2%. Skin, gastrointestinal and eye/ear/mouth infections were of 15.7%, 7.7% and 5.2%. The least common infections were: fever of unknown origin, surgical wound infections and blood infections (2.8%, 2.3% and 1.6%). The same study mentioned that through the analyzes performed on cultures, 253 microorganisms of 36 different types were isolated; the most common microorganisms were *Escherichia coli* (25.7%), *Clostridioides difficile* (13.4%), *Proteus mirabilis* (13%), *Pseudomonas aeruginosa* (7.9%), *Klebsiella pneumoniae* (7.5%), *Staphylococcus aureus* (5.9%) and *Enterococcus faecalis* (3.2%).

According to Salmanov’s et al., [19] study, the most common types of HAI were pneumonia and lower respiratory tract infections (19.4% and 4.1%), surgical wound infections (19.6%), infections of urinary tract (17.5%) and sepsis (10.6%). Of the 232 patients with clinical sepsis, 104 were newborns. In addition, in Salmanov’s study, the prevalence of Gram-negative bacteria was reported, and *Escherichia coli* was the most common pathogen. Gram-positive bacteria were the most common causes of surgical wound infections and sepsis and Gram-negative bacteria were the causes of respiratory tract infections.

The incidence on different types of HAIs varies from country to country and from continent to continent depending on the development of the medical system. For infections associated with ventilator-associated pneumonia, there was a declining tendency in countries with well developed medical systems, e.g., in Germany from 11.2‰ to 8‰, [20], in France from 14.7‰ to 12.6‰ [21], in Korea from 3.48‰ to 1.64‰ [22].

In the case of central line-associated bloodstream infections (CLABSIs) the incidence of CRBSI associated with central lines among patients hospitalized in ICUs in the United States decreased from 3.64 to 1.65 infections per 1000 central-line days between 2001 and 2009 [23,24]. In contrast, the reported pooled incidence of CRBSI across 422 ICUs in 36 countries in Latin America, Asia, Africa, and Europe from 2004 to 2009 was substantially higher, 6.8 events per 1000 central-line days [25].

In the case of catheter associated urinary tract infection (CAUTI), the reported incidence of CAUTI was 3.43/1000 patient-days in EU countries, 3.82/1000 patient-days in non-EU countries [26].

For our cases, it was mentioned in the report sheets that 54.3% of the patients were isolated and 68.6% were in contact with other patients at the time of confirming the case of HAI. Regarding the status at discharge, a quarter of the patients died, a possible cause being the HAI, where despite the fact that a percentage of only 1.1% was mentioned in the reports, we believe that this percentage is underreported. In addition to the cause of death, approximately 2/3 of the cases do not have the mentioned data, but some deaths may be found in this category of data.

One cause for the emergence of HAI is the architecture of the hospitals included in study; in fact, most hospitals in Romania are decades old, the salons sometimes include 6–8 beds, with an increased risk of germ transmission from one patient to another. In addition, there are not many possibilities of isolation of the infectious patients or variants of creating these spaces. Every year, the aspects related to the renovation must be taken into account, because in the hospital, the traffic is intense and the degree of wear of the equipment and of the hospitalization spaces appears.

Given the fact that in Romania, the HAI still remains a much underestimated pathology, it is necessary to implement strategies for improving the activities of surveillance, prevention and limitation of infections in medical units and reducing the causes of non-reporting, such as fear of sanctions; criterion for evaluating the activity of the unit manager; hospital overcrowding; the lack of specialized human resources—epidemiologists and the shortage of trained medical personnel to ensure the prevention and control activities of the HAI; low staff adherence to prevention measures, compliance with recommendations according to guidelines and protocols; lack of specific work procedures or non-application by the medical personnel; the absence of the authority of the epidemiologist/medical director responsible for the decisions of the manager/heads of department/medical personnel; outsourcing microbiology laboratories that serve the hospitals; deficiencies in microbiological diagnosis; improper use of antibiotics [27].

Studying the specialized literature on this topic, we identified a systematic review that established the important elements for organizing effective infection prevention programs in hospitals and key components for monitoring implementation. Ten components were identified as crucial for effective control of hospital infections: organization of infection control at the hospital level; the degree of occupancy of the bed, the personnel employed, the volume of work; availability and easy access to materials and equipment; proper use of guides/protocols/procedures; orientation towards education and lifelong learning; repeated checks; monitoring and feedback; multidisciplinary prevention programs; hiring specialists; positive organizational culture [28]. These components include easy-to-manage and large-scale ways to reduce HAI and improve patient safety.

The plan to limit the infectious risk of a hospital is the practical and applicable expression of an annual infection prevention plan. Adapted to the particularities of each hospital unit, the annual and the multiannual elaborated plan must include measures aimed at ensuring hygiene conditions and observing the strategies of supervision of the HAI to an adequate financing for the modernization and optimization of the activities. Self-assessment of professional practice and clinical audit are powerful tools for increasing the performance of healthcare in terms of effectiveness and efficiency [27].

Education and training of healthcare workers are very important aspects, such as the role of the hands and the surfaces of the environment in the spread of pathogens, the mechanisms of microbial resistance, the correct administration of antimicrobial agents and the precautions for isolation, cannot be ignored today by any medical personnel [29].

Starting from the aspects related to our study of the correspondence between infection, section and pathogen, it is recommended the rapid, vigilant and targeted application of HAI prevention and control strategies for reducing infections in intensive care units [30,31,32,33,34,35], of surgical infections in the pre-, intra- and post-operative phase [36,37,38], reduction of urinary catheter infections [39,40,41], reduction of intravascular catheter infections [42,43], respectively, reduction infections with *Clostridioides difficile* [44].

Limits of the study: Lack of data from the case reports of the HAI, such as the patient’s medical condition upon discharge, or the cause of death in death situations. The epidemiological investigations and the completion of the case reports of the HAI are usually performed within the shortest period from the confirmation of the infection by the laboratory, through the discussions between the epidemiologist and the healing doctor. These infections can occur at any time during the patient’s hospitalization, the report sheets validated by the epidemiologist are transmitted monthly to the County Public Health Directorate, until the 5th of the month for the previous month, so that the supervision, control team of the HAI do not obtain always information about the patient’s condition upon discharge. The fact that the HAI analysis was performed for a county, another limitation of the study is the difference in the prevalence rates of the HAI between the regions of the country, which can affect the national representativeness. However, the 0.44% prevalence rate identified in our study is within the average level mentioned in the few national studies, so we encourage the involvement of several regions in future surveys.

## 5. Conclusions

This study carried out an analysis of the routine prevalence data collected by HAI, showed a clear correspondence between the sections and the type of infections, their associated impact on mortality, which recommends the rapid, vigilant and targeted application of systematic strategies in order to reduce the incidence of HAI. Infections associated with healthcare are a reality, occurring all over the world and their prevention must be a priority for each medical unit and a responsibility for every person involved in healthcare. In the hospital, the infectious risk is ubiquitous, and the development of a management plan for it implies that it is well explained, understood and applied by all staff in a conscious and unconditional way.

## Figures and Tables

**Figure 1 ijerph-17-00760-f001:**
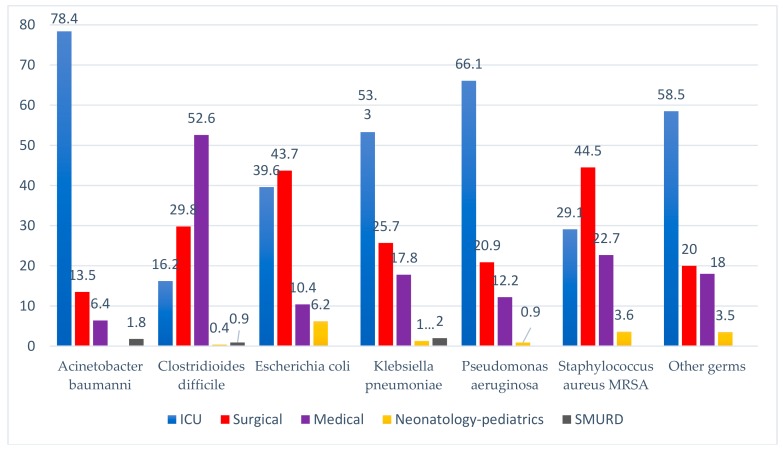
The evidence of the pathogen according to the type of section investigated.

**Figure 2 ijerph-17-00760-f002:**
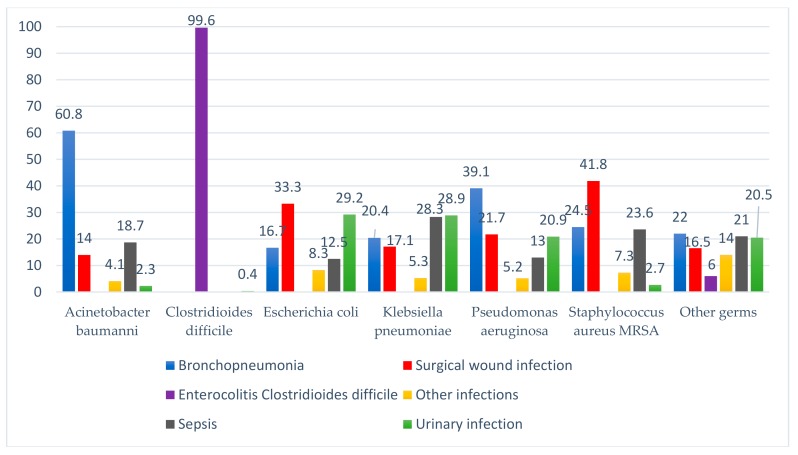
Diagnosis of HAI and type of pathogen.

**Figure 3 ijerph-17-00760-f003:**
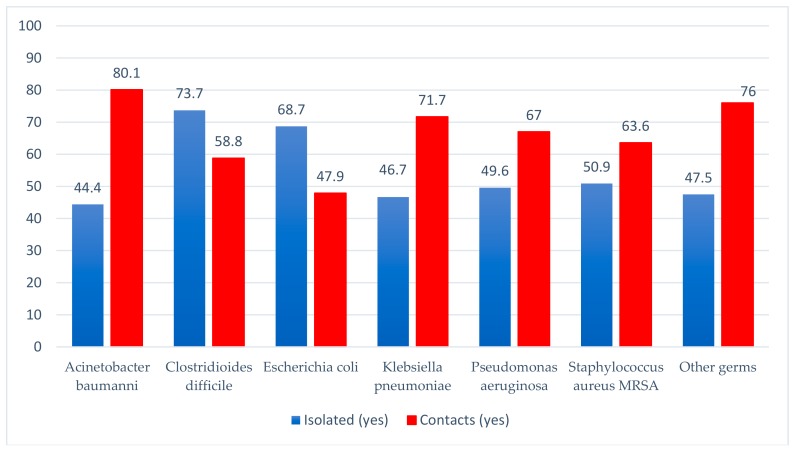
The evidence of the pathogen according to the possibility of isolation of the patient and the existence of contacts.

**Figure 4 ijerph-17-00760-f004:**
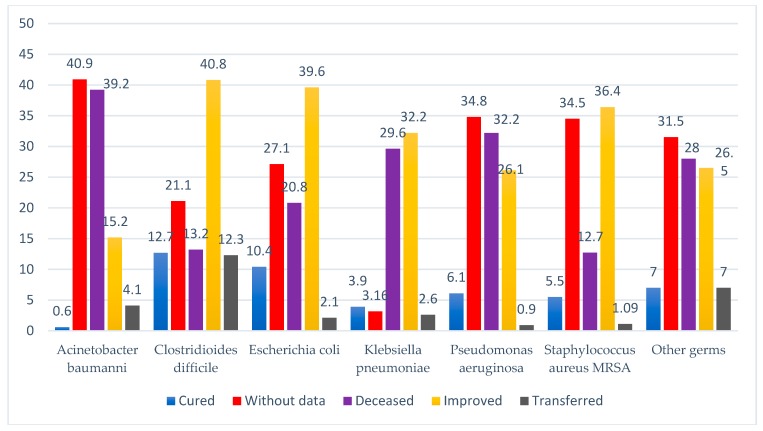
The evidence of the pathogen according the patient’s status upon discharge.

**Table 1 ijerph-17-00760-t001:** Characteristics of the study group.

Caption	Number (1024)	Percentage (%)
Age (mean ± SD)	60.6 ± 19.45	0.1–99 years
Gender (male)	595	58.1
Residence (urban)	542	52.9
**“Section type”**		
Intensive care unit	496	48.4
Surgical	264	25.8
Medical	238	23.2
Pediatric Neonatology	18	1.8
SMURD	8	0.8
**“Infections”**		
Bronchopneumonia	259	25.3
Enterocolitis with *Clostridioides difficile*	239	23.3
Surgical wound infection/soft tissue infection	170	16.6
Sepsis	164	16.0
Urinary infection	131	12.8
Central catheter infection	35	3.4
Influenza type B virus	3	0.3
Other infections	23	2.2
**“Pathogens”**		
*Clostridioides difficile*	228	22.3
*Acinetobacter baumannii*	171	16.7
*Klebsiella pneumoniae*	152	14.8
*Pseudomonas aeruginosa*	115	11.2
*Staphylococcus aureus*	110	10.7
*Escherichia coli*	48	4.7
*Other germs*	200	19.5
**Isolated (yes)**	556	54.3
**Contacts (yes)**	702	68.6
**“Status on discharge”**		
Cured	68	6.6
Transfered	67	6.5
Improved	310	30.3
Deceased	259	25.3
Without data	320	31.3
**“The cause of death”**		
Possibly caused by the HAI	11	1.1
Not related to HAI	219	21.4
Unknown	24	2.3
Without data	770	75.2
**“Origin/classification of the case”**		
From the reporting hospital	888	86.7
From another hospital	91	8.9
Chronic/elderly care units	6	0.6
Other types of medical care	39	3.8

SMURD: “Serviciul Mobil de Urgență, Reanimare și Descarcerare”, which means Mobile Emergency Service for Resuscitation and Extrication.

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
