# Peer review of "Healthcare Associated Infections—A New Pathology in Medical Practice?"

_ijerph, 2020, doi:10.3390/ijerph17030760_

Round 1

Reviewer 1 Report

It is a study about healthcare associated infections (HAIs) in a county in Romania

Thank you for having considered some of the suggestions I wrote in the first review.

I think the Authors have brought some significant improvements to the previous version of the manuscript, especially in the readability of the tables/figures.

However, it is my opinion that it could still benefit of some changes before publication.

Introduction

Page 1, Lines 22-37

The Authors, as suggested, have explained in detail what the Order 1101 was about. Still, I think the description is excessively long. I would suggest to shorten it, focusing only on the differences it brought in comparison with the previous situation.

Material and methods

Page 2, Line 46

I think you should insert into the text of the manuscript the whole explanation about why private hospitals are not included in order to highlight also the differences between private and public structures.

Pages 2-3, Paragraphs 2.2 and 2.3

The description goes too much into the details. I would considerably shorten all this part of the manuscript in a single, brief part, named as 2.2 “Identification and diagnosis of HAIs in Romania”.

Table 1

The addition of the “pathogens” lines is an improvement. I don’t quite understand why you grouped “central catether infections”, “influenza type B virus” together in “other infections”. I would have kept those separated as they were

Results

Figures 1 and 2

It seems that these figures have substituted the former Tables 2 and 3, with considerable improvement in readability. Still, some minor changes are still needed (line breaks for decimal units, numbers overlapping).

Discussion

To have access to the suggested article in extenso [doi: 10.1016/j.jhin.2018.04.006.], I suggest you to ask by e-mail the corresponding author (available at www.doi.org/[10.1016/j.jhin.2018.04.006.]) for the full text article.

Fattorini M., Rosadini D., Messina G., Basagni C., Tinturini A., De Marco MF.

A multidisciplinary educational programme for the management of a carbapenem-resistant Klebsiella pneumoniae outbreak: an Italian experience.

J Hosp Infect. 2018 Apr 6. pii: S0195-6701(18)30218-4.

doi: 10.1016/j.jhin.2018.04.006.

Reviewer 2 Report

The way of data presentation now is much better and more friendly for readers.

There is a mistake in the name "Acinetobacter baumannii". In the whole text there is "baumanni"

On the page 9, line 9 should be "Gram-negative" instead of "gram-negative".

Author Response

This manuscript is a resubmission of an earlier submission. The following is a list of the peer review reports and author responses from that submission.

Round 1

Reviewer 1 Report

This is a study about healthcare associated infections (HAIs) in a county in Romania. 

The results exposed appear quite controversial in comparison with other European countries. The prevalence of HAIs, as the Authors themselves report is extremely low in comparison. This is probably due to the limits that the Authors themselves have expressed.

I would suggest the following changes:

Introduction

Line 62. It is reported only the minimum percentage of 2,6%, it would be necessary to report also the maximum percentage. Line 64. I think it is necessary to better explain in English what the “Order 1101” is about. What changes did it bring or were expected? In what regards? Did it aim to improve the HAIs notifications? How?

Material and methods 

Line 74. I think you should explain why private hospitals were not included Line 84. How was defined if the patient was in contact with other patients? If it was just if the patient was isolated or not, then there’s a repetition. Line 85. How was the antibiotic resistance assessed? Did all the hospitals search only for MRSA? Was this performed in a systematic way?

Results

No mentions of the statistical methods have been run according to what was described in the statistical paragraph of the methods. Results (p values ecc.) of these analyses were not reported in the manuscript. Table 2 and 3. The results are presented as proportions on the total for each column, I think this should be made more evident graphically 

Discussion

The Discussion section is mainly about other studies and report and it is not focused on the results of the study of the Authors. Line 26. The Authors report a “correspondent” (I think it is a typo for “correspondence”?) without any detail, and in the Discussion section (it should be in the Results) Line 72. I’m not sure how these details about the Romanian hospitals’ architecture is relevant to this study, if it was not conducted any analysis differentiating this aspect

I strongly suggest to do a raking with your data (ex frequencies of type of bacteria, type of HAI in each department) with those of other European Countries in order to bypass the scarce incidence that you report due to undereported data.

I suggest the following article witch can enrich your discussion:

Fattorini M., Rosadini D., Messina G., Basagni C., Tinturini A., De Marco MF.

A multidisciplinary educational programme for the management of a carbapenem-resistant Klebsiella pneumoniae outbreak: an Italian experience.

 J Hosp Infect. 2018 Apr 6. pii: S0195-6701(18)30218-4.

doi: 10.1016/j.jhin.2018.04.006.

Reviewer 2 Report

I have some remarks concering the way of result presentation.

But, in the Introduction section there is reference to Order no. 1101 from 2016. I suppose that this is a local Romania document. Please cleary explain it in the introduction.

This document is also cited in Material and Methods. But only general definition of HAI is given. Please provide information about the specific type definitions - are they convergent or based on the ECDC definitions etc. Also, please provide information what was the way of infection detection. Now there is only information "we tracked the infections associated with the medical assistance...". For the readers it is not clear if the cases were detected by the infection control nurse or ward nurse or in another way.

The results section

In my opinion the control row in tables would be useful ("total" for summing for example "Section type" etc.)

Generally, I would recommend to think about a kind of not so detailed way of data presentation as it is now in Table 3 and 4. For example: Table 3, Clostridium difficile and pathogens - 0 for almost all pathogens and 95.0% for CD.  It is obvious, but I would suggest choosing only most often isolated etiological factors.

The same for Table 4. The thought-out, legible, collective way of the results presentation should be adopted in the scientific article.

Some language remarks"

the new name for Clostridium difficile is Clostridioides difficile. The name of species in the article are not always written using italic font - it should be corrected.
